# Reliability of Computing van der Waals Bond Lengths of Some Rare Gas Diatomics

**DOI:** 10.3390/ijms232213944

**Published:** 2022-11-11

**Authors:** Yi-Liang Zhang, Bin Li

**Affiliations:** College of Chemistry, Jilin University, Qianjin Street 2699, Changchun 130012, China

**Keywords:** density-functional theory (DFT) methods, ab initio computations, van der Waals molecules, bond length, CCSD(T) method, computational chemistry

## Abstract

When the bond lengths of 11 molecules containing van der Waals bonds are optimized by 572 methods and 20 basis sets, it is found that the best mean absolute deviations (MADs) of density-functional theory (DFT) methods are 0.005 Å (shown by APFD/6-311++G**), 0.007 Å (B2PLYPD3(Full)/aug-cc-pVQZ), and 0.010 Å (revDSDPBEP86/aug-cc-pVQZ), while the best MADs of ab initio methods are 0.008 Å (BD(T)/aug-cc-pVTZ) and 0.016 Å (MP4/aug-cc-pVQZ). Moreover, the best MADs calculated by 54 selected methods in combination with 60 other basis sets (such as 6-311++G, 6-31++G(3d′f,3p′d), and UGBS1V++) are not better. Therefore, these bond lengths can be calculated with extremely high accuracy by some special methods and basis sets, and CCSD(T) is also not as good as expected because its best MAD is only 0.023 Å (CCSD(T)/aug-cc-pVQZ).

## 1. Introduction

It is well known that experiment, theory, and computation are the three pillars of modern chemistry [1]. Like natural science, where experiment is the driving force for the establishment, testing, and development of a science, the core of chemistry is also experiment, such as synthetic chemistry, which has reached the stage of synthesizing any complexity (e.g., palytoxin) and any number (e.g., 200 million) of molecules. From experimental chemistry has developed theoretical chemistry making chemistry from empirical to rational, and theoretical chemistry becomes more and more important. As the center and the basics of theoretical chemistry, quantum chemistry was born when Heitler and London used quantum mechanics to explain why two neutral hydrogen atoms could form a chemical bond in 1927. It can be said that it is a science of describing chemical bonds at the molecular level. Therefore, chemical bonds are the core, the essence, and the golden key of chemistry. The developed result of theoretical chemistry is computational chemistry, which is also the executor and extension of theoretical chemistry, since computation is the bridge to connect theoretical models and experimental results. The development of chemical theories is inseparable from chemical experiments, although they can also be used to guide experiments. Computational chemistry combines theory and experiment, unites computer and chemistry, and uses computers as experimental tools. Theoretical chemistry and computational chemistry make chemistry more rigorous, quantitative, and mature, so that chemistry is no longer a purely experimental science.

The essence of chemistry is chemical reaction or chemical change, which is the breaking and forming of chemical bonds. It is also generally believed that structure determines everything, that is, molecular structure determines molecular properties, and molecular properties determine the chemical properties of a substance, so the properties of the substance are mainly determined by its molecular structure (e.g., sometimes external factors such as solvents are also important). However, a molecule is held together by chemical bond(s), or its structure depends on its chemical bond(s). That is to say, properties are the representation mode of structures, and structures are determinative. On the other hand, chemical bonds are the essence, and structures are their manifestation. Therefore, chemical bonds determine structures and properties by reasoning.

Chemical bonds are the attractions that join atoms in a compound, resulting in lower energy and greater stability. Although Pauling believed that Xenon might take place chemical reactions in 1933, it was not until Bartlett synthesized the first rare gas compound XePtF_6_ in 1962 [2] that the previous statement that rare gases could not participate in reactions was overturned. In 1873, van der Waals first discovered noncovalent interactions that helped him to reformulate the equation of state for real gases [3,4]. London used quantum mechanics to calculate dispersion forces in 1930 [5]. Bonding is inherently quantum mechanical in nature, so only quantum mechanics can provide valid theoretical tools to understand the nature of chemical bonds. For instance, the binding force of chemical bonds is formed by the wave action of electrons, or due to the special wave-particle duality, it is no wonder that it goes beyond the boundary of “same charges repel, different charges attract”. It is now recognized that it is convenient to classify chemical bonds into six types: covalent bond, ionic bond, coordination bond, metallic bond, hydrogen bond, and van der Waals bond. Similarly, the interactions between atoms can also be classified into six categories:
①Two closed-shell neutral atoms, e.g., He-He, or a van der Waals bond;②Two open-shell neutral atoms, e.g., H-H, or a covalent bond in which two atoms share electron pair(s);③One closed-shell positive ion and one closed-shell negative ion, e.g., Na^+^−Cl^−^, or an ionic bond of an electrostatic interaction;④One closed-shell atom or molecule with lone pair(s) of electrons and one or more open-shell atoms (including partially positively charged hydrogen(s) when bonded to an electronegative element) or molecules, e.g., a N−H·O hydrogen bond between oxygen and hydrogen, and a dative B−N bond or a special kind of covalent bonds in which one atom donates two electrons;⑤One open-shell ion (usually a positive ion) and one or several closed-shell ions or molecules, e.g., M^n+^(X^−^)_m_, or a coordination bond;⑥Many metal atoms aggregate together in which outer shell electrons break away from the core and move around all metal solids, e.g., a metallic bond.

The properties of chemical bonds are characterized by bond parameters such as bond length, bond angle, bond energy, bond order, and dipole moment, which can be measured by experiments directly or indirectly, or obtained from theoretical computations. Although direct experimental assessment of van der Waals interactions is intrinsically difficult [6], the notion that van der Waals bonds do exist can be seen from their very precisely measured bond lengths and bond energies. For example, the measured equilibrium bond length (*r*_e_) and bond dissociation energy of NeKr are, respectively, 3.645444(13) Å [7] and 4.31 kJ/mol [8]. It is then doubtful whether computational chemistry can obtain these values theoretically and accurately, since theories are now usually considered to have achieved experimental accuracy. However, some popular density-functional theory (DFT) methods such as B3LYP and PBE1PBE have been found to be very poor in obtaining van der Waals bond lengths, and the “gold standard” [9] CCSD(T) is also not as good as expected when calculating molecular structures [10]. Additionally, van der Waals diatomics have not been very systematically studied in the references [11,12,13,14], so a more in-depth study is necessary, which is the main focus of this work.

## 2. Results and Discussion

### 2.1. Computational Results of 572 Methods and 20 Basis Sets

First, 11 rare gas molecules (diatomics) containing van der Waals bonds, represented below by 11-RG-Mols, are optimized by 572 methods and 20 basis sets. These methods include almost all DFT and ab initio methods known to us. Among these 20 basis sets, there are four of Ahlrichs’ [15], eight of Pople’s [16,17,18], and eight of Dunning’s [19,20] as they perform relatively better, are more commonly used, and can be used to calculate all these 11-RG-Mols. These 20 basis sets are numbered from BS01 to BS20 and are listed in Appendix A. The relative reliability of a method/basis set is based on its mean absolute deviation (MAD) value when computing the 11-RG-Mols. All their MADs are listed in Appendix A (their mean deviations (MDs) are also listed in Appendix A). It is then found that the best MADs (0.005 Å, 0.006 Å, 0.007 Å, 0.008 Å, and 0.008 Å) are shown by the same DFT methods as in ref. [10] (i.e., APFD [14], B2PLYPD3 [21,22,23], and B2PLYPD3(Full)) in combination with several basis sets, i.e., shown respectively by APFD/6-311++G**, APFD/6-311++G(df,pd), B2PLYPD3(Full)/aug-cc-pVQZ, B2PLYPD3/aug-cc-pVQZ, and B2PLYPD3/aug-cc-pV5Z. The same best result of 0.008 Å is also shown by BD(T) [24], i.e., BD(T)/aug-cc-pVTZ, while the other best MADs of ab initio methods are 0.012 Å, 0.013 Å, 0.016 Å, and 0.016 Å, shown respectively by BD(T)(Full)/aug-cc-pVQZ, BD(T)/aug-cc-pVQZ, BD(T)/aug-cc-pV5Z, and MP4/aug-cc-pVQZ (i.e., other Møller-Plesset perturbation methods such as MP2, MP3, MP3(Full), MP4(Full), and MP5, whose MADs in combination with aug-cc-pVQZ are, respectively, 0.061 Å, 0.034 Å, 0.022 Å, 0.025 Å, and 0.027 Å are poorer). These data, and some other better or special methods, are listed in Table 1. The worst MAD (as large as 4.074 Å) is shown by G96B95/aug-cc-pVDZ. Unless otherwise stated, results of PM7, PM6, and UFF are not counted.

The methods which may be considered to be “accurate” (i.e., with MAD(s) between 0.010 Å and 0.020 Å) [25] or better (i.e., with MAD(s) even smaller than 0.010 Å) are shown by APFD [14], B2PLYPD3 [21,22,23], B2PLYPD3(Full), DSDPBEP86 [26,27], DSDPBEP86(Full), revDSDPBEP86 [28], revDSDPBEP86(Full), BD [24], BD(Full), MP4 [29,30], BD(T) [24], and BD(T)(Full). All these 12 methods are listed in Table 1. Their total number of method/basis sets is 33, its percentage relative to these 12 × 20 or 240 method/basis sets is 13.75%, and its percentage relative to all calculated 11436 method/basis sets is as low as 0.29%. Moreover, if the number (i.e., 792) of method/basis sets with MADs less than 0.100 Å (the MAD of UFF is 0.094 Å, also less than 0.100 Å) is counted, its percentage of 6.93% is also small. That is, most method/basis sets are very unreliable in computing these 11-RG-Mols. Methods with at least 10 times (i.e., at least 10 basis sets combined with a method with MADs less than 0.100 Å) include: 10 (BD(T)(Full), B2PLYPD, and M06-D3), 11 (PBEB95, LC-XaB95, and PBE1PBE-D3), 12 (LC-BRxB95, B2PLYP-D3, B2PLYP-D3(Full), and PBE1PBE-D3BJ), 13 (LGKCIS), 14 (wB97X, M06, LGB95, LC-G96B95, LC-RevTPSSB95, LC-TPSSB95, B2PLYPD3, B2PLYPD3(Full), and B3LYP-D3), 15 (LC-BB95, LC-LGB95, LC-mPWB95, LC-OB95, LC-PBEB95, LC-PBEhB95, LC-PKZBB95, LC-PW91B95, and LC-SB95), 16 (APFD and MN15L), 17 (MN15, DSDPBEP86, DSDPBEP86(Full), revDSDPBEP86, and revDSDPBEP86(Full)), and 18 times (PW6B95D3), which are all DFT methods except BD(T)(Full), e.g., the times of MP4(Full), CCSD(T)(Full), and QCISD(T)(Full) are all only 8.

In Table 2, the best and next best methods for these 20 basis sets are listed. It can be seen that their best MADs vary widely between 0.005 Å and 0.064 Å. The number of methods whose best MADs do not exceed 0.020 Å is 11. For these best MADs, the largest time is 8, shown by APFD, while the next largest time is 3, shown by DSDPBEP86. The other methods with 2 times include DSDPBEP86(Full), B2PLYPD3(Full), M06, and BD(T). For the next best methods, the largest times is 6, shown by DSDPBEP86(Full). For the basis sets whose best and next best MADs are both accurate or even better which are shown by methods of APFD, B2PLYPD3, B2PLYPD3(Full), DSDPBEP86, DSDPBEP86(Full), revDSDPBEP86(Full), and/or BD(T), only five Dunning’s basis sets can satisfy this requirement, i.e., aug-cc-pVTZ, cc-pVQZ, aug-cc-pVQZ, cc-pV5Z, and aug-cc-pV5Z. To obtain accurate or better bond lengths of the 11-RG-Mols, some special methods and basis sets must be used. As found before, diffuse functions are often important for calculating these 11-RG-Mols, e.g., the best MADs for 6-311G**, cc-pVDZ, and cc-pVTZ are, respectively, 0.064 Å, 0.056 Å, and 0.051 Å, while those for 6-311++G**, aug-cc-pVDZ, and aug-cc-pVTZ are, respectively, 0.005 Å, 0.025 Å, and 0.008 Å, which are much better. However, the situation is somewhat different for cc-pVQZ (with the best MAD of 0.016 Å), cc-pV5Z (0.012 Å), and Def2QZVP (0.012 Å).

The frequencies of these 36 methods listed in Table 1 in combination with all the 20 basis sets are also computed with their corresponding optimized bond lengths to see whether there are imaginary frequencies. An imaginary or a negative frequency means that this molecule is unstable (e.g., a transition state) judged from this method/basis set. As can be seen from their detailed data listed in Appendix A, imaginary frequencies appear for all these 11-RG-Mols, but the number for one molecule ranges from 1 for ^20^Ne^84^Kr to 62 for ^4^He_2_. The total number of method/basis sets with imaginary frequency is 166, and its percentage relative to 7920 is only 2.10%, which is negligible, so nearly all geometries optimized by the corresponding method/basis sets are judged to be energy minima or stable conformations, meaning that these calculations can be considered successful. However, as listed in the Supporting Information of ref. [10], the number with imaginary frequency of 289 problematic molecules (almost all non-aromatic) is 69 when calculated by 26 method/basis sets (excluding PM6 and UFF results), and that of the total calculated 7514 method/basis sets is 741. The percentages of 69 versus 289 and 741 versus 7514 are, respectively, as high as 23.88% and 9.86%. It has been found that the number of imaginary frequencies which will exclude the stability of these two molecules is 1125, or its percentage relative to the total number of computed method/basis sets is 18.18% when planar benzene and nonplanar toluene are computed by more than one hundred methods and more than one hundred basis sets [31]. Therefore, the reason may be similar to calculating vibrational frequencies of aromatic molecules, i.e., it is also impossible to judge the stability of some non-aromatic molecules, including these van der Waals ones when using many method/basis sets of computational chemistry.

### 2.2. Computational Results of 54 Methods and 60 Other Basis Sets

Next, 54 relatively better methods are selected to optimize these 11-RG-Mols in combination with 60 other basis sets, including nearly all types for which they can be computed (e.g., STO-6G, TZVP, LanL2DZ, 6-31+G, CBSB7, 6-31++G(2df,2pd), 6-31++G(3d’,3p’), and UGBS), and their MADs are listed in Appendix A (their MDs are listed in Appendix A). These 60 basis sets are numbered from BS21 to BS80, also listed in Appendix A. If these DFT methods combined with UGBS and UGBS1V++ which can calculate only three molecules (i.e., like Dreiding, they can only calculate ^4^He_2_, Ne_2_, and HeNe) are not considered, it is seen that methods with “accurate” MADs only include BD(T), BD(T)(Full), and APFD, i.e., shown by BD(T)/6-31++G(3d,3p) (with MAD of 0.018 Å), BD(T)/6-31++G(3d’,3p’) (0.020 Å), BD(T)(Full)/6-31++G(3d’f,3p’d) (0.017 Å), APFD/6-311++G (0.017 Å), APFD/6-311++G* (0.011 Å), and APFD/6-311++G(df) (0.011 Å). Then, when they are used to optimize the 11-RG-Mols with APFD, the results of 6-311++G, 6-311++G*, and 6-311++G(df) are slightly worse than those of 6-311++G** and 6-311++G(df,pd) in Section 2.1, and other similar basis sets such as 6-31G* (with MAD of 0.354 Å), 6-31+G* (0.149 Å), 6-31++G** (0.108 Å), 6-31++G(3df,3pd) (0.173 Å), 6-311+G (0.034 Å), 6-311+G* (0.030 Å), 6-311+G** (0.049 Å), and CBSB7++ (0.028 Å) can be much worse. The worst MAD is 1.196 Å shown by MP3/CEP-31G or MP3(Full)/CEP-31G. The number of MADs smaller than 0.100 Å is 588, or its percentage relative to 3084 is 19.07%, much better than the overall 6.93% in Section 2.1. However, if only these 54 methods whose number with MAD smaller than 0.100 Å is 514 in Section 2.1 are counted, their percentage of 47.87% is much larger than this 19.07%, so the basis sets in Section 2.1 are generally better. It is noted that the number with MAD smaller than 0.100 Å of PW6B95D3 (with a maximum number of 18 for the former 20 basis sets) or APFD in combination with these 60 basis sets is 26, which is smaller than the maximum of 39 shown by M06, but as listed above, none of their MADs does not exceed 0.020 Å, except for six basis sets combined with APFD, BD(T), and BD(T)(Full).

The results of CCSD(T) [32] should be pointed out because it is generally considered as the only widely accepted gold-standard method. In Section 2.1, its best MAD is 0.023 Å, shown by CCSD(T)/aug-cc-pVQZ, as shown in ref. [10], while in this Section its best MAD is 0.026 Å, shown by CCSD(T)(Full)/6-31++G(3d’,3p’) (similar results for QCISD(T)). Therefore, the two best MADs are not accurate, nor better than revDSDPBEP86, BD(T), MP4, not to mention such DFT methods as APFD and B2PLYPD3(Full). In ref. [10], it is also found that not only are the best MADs of bond lengths (i.e., 0.021 Å shown by DSDPBEP86(Full)/Def2QZVP and MP2(Full)/cc-pVQZ) and bond angles (i.e., 2.3° shown by QCISD(T)/cc-pVTZ) not shown by CCSD(T), but also the best SUM1 (5.0 unit) and SUM2 (2.5 unit) of CCSD(T) shown by CCSD(T)/cc-pVTZ and CCSD(T)/6-311++G(3df,3pd) (shown also by QCISD(T)/cc-pVTZ and CCSD(Full)/cc-pVQZ for these same SUM1 and SUM2, with SUM1 and SUM2 measuring its overall reliability of a method/basis set calculating both bond length and bond angle) are larger than those (4.9 unit and 2.4 unit) of DSDPBEP86/cc-pVQZ, DSDPBEP86(Full)/cc-pVQZ, DSDPBEP86/aug-cc-pVQZ, DSDPBEP86/6-311++G(3df,3pd), DSDPBEP86(Full)/6-311++G(3df,3pd), and MP2/aug-cc-pVQZ. Therefore, it is no longer appropriate to call CCSD(T) the “gold standard” for the purpose of calculating molecular structures (at least for the lengths and angles of covalent and ionic bonds, and the lengths of van der Waals bonds).

Then, from this study and ref. [10], it is best to use DFT methods such as APFD, B2PLYPD3(Full), DSDPBEP86(Full), and revDSDPBEP86 to obtain better structures. For example, in ref. [10], it was found that DSDPBEP86, DSDPBEP86(Full), and APFD all performed very well (meaning that their SUM1s do not exceed 5.4 units and their SUM2s do not exceed 2.6 units) when used to optimize 179 molecules in combination with 6 basis sets (i.e., cc-pVTZ, cc-pVQZ, aug-cc-pVQZ, 6-311++G(3df,3pd), Def2TZVPP, and Def2QZVP) (this criterion is also met by 6 other DFT methods, i.e., OHSE1PBE, SOGGA11X, PBE1PBE, PBE1PBE-D3, PBE1PBE-D3BJ, and B3PW91-D3BJ). Although the number of these 6 basis sets which behave very well is only 4 for B2PLYPD3(Full), it is 2 for MP2 and CCSD(T). Not only are their computations (such as consumed memories and CPU times) much easier than CCSD(T), but also they are not inferior to CCSD(T), and can even be better.

## 3. Materials and Methods

These 11-RG-Mols include ^4^He_2_, Ne_2_, ^40^Ar_2_, ^84^Kr_2_, HeNe, HeAr, HeKr, ^20^Ne^40^Ar, ^22^Ne^36^Ar, ^20^Ne^84^Kr, and ^40^Ar^84^Kr, whose spectroscopic states are all $X^{1}\sum {}_{g}^{+}$ or X^1^∑^+^, and whose adopted experimental equilibrium bond lengths (*r*_e_) [7,11,12,13,33] are, respectively, 2.970 Å, 3.091 Å, 3.758 Å, 4.017(12) Å, 3.05 Å, 3.50 Å, 3.69 Å, 3.46360(12) Å, 3.46539(55) Å, 3.645444(13) Å, and 3.894358(50) Å, also listed in ref. [10]. As in ref. [10], their experimental bond lengths are usually used as their initial parameters to be optimized. These theoretically computed bond lengths are then compared with their corresponding experimental ones and the MAD ($\mathrm{MAD}=(1/11)\sum_{i=1}^{11}\left|r_{i,\mathrm{calc}}-r_{i,\exp}\right|$ [34]) of a method/basis set is obtained. These calculations are carried out by using Gaussian 16 program [35].

## 4. Conclusions

After the computations of the 11-RG-Mols containing van der Waals bonds with 572 methods and 80 basis sets, it is found that the best or next best MADs are usually shown by such methods as APFD, B2PLYPD3(Full), DSDPBEP86, and DSDPBEP86(Full), which are also very good for calculating the 179 problematic molecules with covalent and/or ionic bonds [10]. That is to say, to obtain very good structures for these three bond types, it is best to choose the above DFT methods in combination with some large or very large basis sets. CCSD(T) results do not meet the criterion of accurate bond lengths, so it is not suitable to be used as a comparing standard, as other ab initio methods like BD(T) and MP4 can be better than CCSD(T) in calculating these van der Waals bond lengths.

## Figures and Tables

**Table 1 ijms-23-13944-t001:** MADs (Å) of some methods calculating the 11-RG-Mols.

	D2TP	D2QPP	++dp	++3d	++df	++3df	aug-T	aug-Q	cc-5	aug-5	Ave
HF	0.791	1.309	0.914	0.726	0.913	0.726	1.105	1.259	1.308	1.308	1.036
MP2	0.219	0.198	0.273	0.091	0.241	0.076	0.057	0.061	0.153	0.063	0.143
B2PLYPD3	0.038	0.053	0.183	0.099	0.155	0.095	0.014	0.008	0.024	0.008	0.068
B2PLYPD3(Full)	0.037	0.038	0.179	0.085	0.154	0.083	0.020	0.007	0.012	0.010	0.063
DSDPBEP86	0.018	0.035	0.052	0.058	0.048	0.060	0.028	0.026	0.013	0.029	0.037
DSDPBEP86(Full)	0.018	0.025	0.050	0.061	0.040	0.066	0.072	0.041	0.014	0.056	0.044
revDSDPBEP86	0.032	0.078	0.067	0.055	0.058	0.055	0.015	0.010	0.034	0.017	0.042
revDSDPBEP86(Full)	0.023	0.059	0.064	0.060	0.054	0.060	0.049	0.019	0.022	0.033	0.044
CCSD(Full)	0.245	0.225	0.274	0.092	0.254	0.078	0.038	0.025	0.142	0.033	0.140
BD	0.249	0.215	0.280	0.090	0.241	0.071	0.030	0.019	0.097	0.022	0.131
BD(Full)	0.249	0.212	0.280	0.090	0.241	0.071	0.030	0.019	0.097	0.022	0.131
MP4	0.244	0.203	0.262	0.098	0.225	0.071	0.031	0.016	0.115	0.027	0.129
CCSD(T)	0.250	0.167	0.260	0.098	0.226	0.072	0.030	0.023	0.107	0.026	0.126
CCSD(T)(Full)	0.201	0.166	0.244	0.056	0.204	0.049	0.060	0.025	0.083	0.060	0.115
BD(T)	0.207	0.150	0.224	0.048	0.187	0.038	0.008	0.013	0.098	0.016	0.099
BD(T)(Full)	0.180	0.178	0.235	0.066	0.271	0.043	0.038	0.012	0.074	0.027	0.112
APFD	0.094	0.012	0.005	0.054	0.006	0.056	0.023	0.012	0.012	0.013	0.029
wB97X	0.102	0.065	0.082	0.076	0.083	0.077	0.070	0.051	0.067	0.078	0.075
B3LYP	0.982	1.910	1.416	1.069	1.380	1.074	1.772	1.910	1.828	1.990	1.533
PBE1PBE	0.148	0.155	0.148	0.122	0.142	0.121	0.147	0.154	0.153	0.153	0.144
M06	0.065	0.141	0.072	0.070	0.072	0.070	0.068	0.170	0.140	0.152	0.102
M06L	0.121	0.140	0.127	0.074	0.127	0.073	0.066	0.099	0.110	0.076	0.101
MN15	0.094	0.081	0.075	0.066	0.075	0.066	0.066	0.072	0.082	0.088	0.076
MN15L	0.072	0.086	0.076	0.070	0.076	0.062	0.061	0.081	0.086	0.089	0.076
PW6B95D3	0.081	0.089	0.082	0.069	0.082	0.069	0.084	0.088	0.089	0.090	0.082
PBEB95	0.119	0.109	0.057	0.065	0.057	0.065	0.084	0.092	0.109	0.097	0.086
LC-PBEB95	0.060	0.099	0.086	0.084	0.086	0.084	0.086	0.096	0.099	0.100	0.088
B3LYP-D2	0.170	0.097	0.093	0.110	0.094	0.111	0.091	0.100	0.100	0.097	0.106
B3LYP-D3	0.106	0.064	0.068	0.087	0.069	0.087	0.043	0.065	0.071	0.066	0.073
M06-D3	0.068	0.172	0.136	0.073	0.136	0.073	0.123	0.172	0.171	0.118	0.124
M06L-D3	0.122	0.141	0.128	0.098	0.128	0.074	0.067	0.101	0.111	0.128	0.110
LC-wPBE-D3	0.049	0.157	0.162	0.115	0.161	0.112	0.140	0.173	0.155	0.166	0.139
PBE1PBE-D3	0.121	0.089	0.096	0.111	0.097	0.112	0.084	0.090	0.092	0.088	0.098
PBE1PBE-D3BJ	0.112	0.087	0.084	0.102	0.084	0.102	0.082	0.090	0.086	0.087	0.091
PBEPBE-D3BJ	0.150	0.097	0.098	0.118	0.098	0.119	0.090	0.095	0.104	0.097	0.107
B3LYP-D3BJ	0.052	0.083	0.165	0.150	0.165	0.154	0.151	0.077	0.073	0.164	0.123
Ave	0.164	0.200	0.197	0.129	0.187	0.124	0.140	0.149	0.170	0.158	0.162

The D2TP, D2QPP, ++dp, ++3d, ++df, ++3df, aug-T, aug-Q, cc-5, and aug-5 denote, respectively, Def2TZVP, Def2QZVPP, 6-311++G**, 6-311++G(3d,3p), 6-311++G(df,pd), 6-311++G(3df,3pd), aug-cc-pVTZ, aug-cc-pVQZ, cc-pV5Z, and aug-cc-pV5Z. Ave is abbreviation of average for these considered 36 methods or 10 basis sets. A green or blue value means, respectively, that this MAD is smaller than 0.020 Å or between 0.020~0.100 Å, while a black one means that this value is larger than 0.100 Å.

**Table 2 ijms-23-13944-t002:** The best and next best MADs (Å) and methods for the 20 basis sets.

Basis Set	The Best		The Next Best	
	MAD	Method	MAD	Method
Def2TZVP	0.018	DSDPBEP86, or DSDPBEP86(Full)	0.023	revDSDPBEP86(Full)
Def2TZVPP	0.027	DSDPBEP86(Full)	0.029	DSDPBEP86
Def2QZVP	0.012	APFD	0.029	DSDPBEP86(Full)
Def2QZVPP	0.012	APFD	0.025	DSDPBEP86(Full)
6-311G**	0.064	M06	0.069	M06-D3
6-311++G**	0.005	APFD	0.049	wPBEhB95
6-311++G(2d,2p)	0.036	DSDPBEP86	0.040	DSDPBEP86(Full)
6-311++G(3d,3p)	0.054	APFD	0.055	revDSDPBEP86
6-311++G(df,pd)	0.006	APFD	0.040	DSDPBEP86(Full)
6-311++G(2df,2pd)	0.036	revDSDPBEP86(Full)	0.038	DSDPBEP86
6-311++G(3df,3pd)	0.038	BD(T)	0.043	BD(T)(Full)
6-311++G(3d2f,3p2d)	0.012	APFD	0.061	PBEB95
cc-pVDZ	0.055	M06HF	0.056	M06HF-D3
aug-cc-pVDZ	0.025	APFD	0.036	DSDPBEP86(Full)
cc-pVTZ	0.051	M06	0.053	M06-D3
aug-cc-pVTZ	0.008	BD(T)	0.014	B2PLYPD3
cc-pVQZ	0.016	DSDPBEP86	0.018	DSDPBEP86(Full), or revDSDPBEP86(Full)
aug-cc-pVQZ	0.007	B2PLYPD3(Full)	0.008	B2PLYPD3
cc-pV5Z	0.012	APFD, or B2PLYPD3(Full)	0.013	DSDPBEP86
aug-cc-pV5Z	0.008	B2PLYPD3	0.010	B2PLYPD3(Full)

As in Table 1, a green or blue value means, respectively, that this MAD is smaller than 0.020 Å or between 0.020~0.100 Å. , while a black one means that this value is larger than 0.100 Å.

## Data Availability

The data that support this study are available in the Appendix A and from the corresponding author upon reasonable request.

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
