# Peer review of "Reliability of Computing van der Waals Bond Lengths of Some Rare Gas Diatomics"

_ijms, 2022, doi:10.3390/ijms232213944_

Round 1

Reviewer 1 Report

This is a valuable study complementing Zhang, Wang and Ren (2022) (ref 10 in the manuscript) by focussing on Van der Waals bonded diatomics. A very large number of method/basis-set combinations has been used and the authors do a good job in digesting this information to essentials.

I have some critical comments.

The introduction looks inappropriate to me for this article; it is much to broad and the text would serve better in the introduction to a textbook or to a quite general review article. For the present narrow study the introduction could be cut down to a little more than just its present final paragraph.

The concept of experimental bond length needs to be defined with respect to nuclear zero-point motion.

The discussion of imaginary frequencies at the end of Sec 3.1 is unclear to me. If there is an imaginary frequency then it means that the configuration is unstable with respect to nuclear displacements; it is not a local minimum for the electronic state that is considered. I don't know why electronic excitation would bring it to a lower energy state. If any of the method/basis combinations in Table I or Table II show an imaginary frequency then that had better be shown in the table. (E.g., attach an asterisk to the entry.) If none of those entries have an imaginary frequency then it is good to say that explicitly.

The results for the sequence of 6-311++G basis sets in Table II seem to be jumping up and down without much reason; i.e., expanding the basis may give a worse result. At first this raises the concern that the results just reflect an accidental match between method and basis set for the particular 11 molecules in this study. On the other hand, it does seem that the same methods (esp APFD) keep showing up as the best. Maybe the authors can say something more about the seemingly erratic behaviour of accuracy vs basis set for the same method.

Author Response

Thanks very much for your good comments!

I agree that most contents of the introduction are more suitable to be put in a textbook or review. However, in the books and reviews known by me, their answers to most of my questions cannot be found. For example, in the popular textbooks on physical chemistry and quantum chemistry (even on non-covalent interactions), although van der Waals molecules are introduced (e.g., I. N. Levine: Quantum Chemistry, 2014, p. 374; P. Atkins, J. de Paula: Atkins Physical Chemistry, 2002, p. 709; I. N. Levine: Physical Chemistry, 2009, pp. 865−866; A.O. de la Roza, G.A. DiLabio: Non-Covalent Interactions in Quantum Chemistry and Physics, 2017; S. Scheiner, Ed.: Noncovalent Forces, 2015), their essences are not written or are not clear. So it is necessary to point out how they are different from other bond types, then the third paragraph is necessary in which the six different categories of bonds have been differentiated and they are extended and summarized by me from known knowledge. To go further, the position of chemical bonds in chemistry and in computational chemistry had better be simultaneously introduced so that one can have a comprehensive realization to them which is seen in the first and second paragraphs. In the introduction one can review a topic and the first two paragraphs are not long so they are also retained.

The optimized theoretical bond length re corresponds to the equilibrium structure of an isolated molecule at 0 K and at rest, while the experimental re is the distance between equilibrium nuclear positions and refers to averages over zero-point vibrational motion. They are all equilibrium bond lengths and can be directly compared, which is also used by a lot of references. Moreover, their experimental re are usually very precise, e.g., the re of 40Ar84Kr (X1+) is 3.645444(13) Å, while the combined effects of theoretical and experimental uncertainties render comparisons dubious below the level of 0.01 Å for lengths (H. F. Schaefer III, Ed.: Applications of Electronic Structure Theory: Modern Theoretical Chemistry, Vol. 4, 1977, p. 7) which is much larger than its uncertainty or error.

The book (original ref. [34]) saying this viewpoint is deleted, because it does not tell its reason clearly. Every datum in Table 1 or Table 2 is the MAD considering 11 computed bond lengths. However, the maximum number showing imaginary frequencies for all these 11 molecules is only 5 (shown by PBEB95/aug-cc-pVDZ), and others are e.g. 4 (shown only by BD(T)(Full)/6-311++G(3d2f,3p2d) and PBEB95/6-311++G(3d2f,3p2d)) and 3 (shown only by PBEB95/6-311++G**, PBEB95/6-311++G(df,pd), LC-PBEB95/Def2TZVP, and LC-PBEB95/Def2TZVPP), so none of the method/basis sets shows 11 imaginary frequencies for all these 11 molecules and none is suitable to be attached an asterisk. They are usually accidental and are consistent with the viewpoint that their overall percentage is negligible.

To my knowledge, 352 different basis sets can be combined when using Pople’s k-nlmG type that is too many. For instance, the basis sets changing only one place relative to 6-311++G** include 6-311++G*, 6-311+G**, 6-31++G**, 6-311++G(d,2p), 6-311++G(d,3p), 6-311++G(d,pd), 6-311++G(d,2pd), 6-311++G(d,3pd), 6-311++G(d,3p2d), 6-311++G(2d,p), 6-311++G(3d,p), 6-311++G(df,p), 6-311++G(2df,p), 6-311++G(3df,p), and 6-311++G(3d2f,p), some of which are also explored in Section 3.2. One of the aims for this work is to seek the basis sets and methods that perform the best. However, except for APFD whose best results appear in combination with 6-311++G** and 6-311++G(df,pd), the best basis sets for other comparatively better methods are mostly Dunning’s cc-pVnZ ones (especially aug-cc-pVQZ, e.g., revDSDPBEP86, MP4, and CCSD(T)) which are all calculated since cc-pV6Z cannot be used to compute all these 11 molecules. It is also seen in Table 2 or line 168 that all the five basis sets whose best and second best MADs are both accurate or even better are Dunning’s.

Reviewer 2 Report

The paper described the reliability of a large number (572) of DFT methods against initially 20 basis sets of 11 rare gas diatomics containing vdW bonds. Using the top performing methods (54), 60 additional basis sets were tested.  The results were observed to be consistent with the results of another paper for 1342 molecules using 30 methods and 7 basis sets.  The paper is very interesting and I recommend publication after some minor revisions.

- The introduction can be improved, for example, discussing and citing papers that can clearly demonstrate the problem/need to carry out the study in the manuscript.

- The english writing style can also be improved to improve the flow of discussion especially in the introduction and discussion.

- It is quite hard to follow fully tabulated results especially for large data sets such as the one presented in the study.  Probably the authors can make plots/graphs, e.g., scatter plots, clustering, histograms, etc., from the results for better visualization and easier understanding.

Author Response

Thank for your affirmation!

The several cited papers (i.e., the original refs. [12], [13], [14], and [23]) discussed the calculation and comparison of van der Waals diatomics, so are shifted to the introduction (i.e., refs. [11]−[14]).

The writing of this manuscript has been examined by an English-speaking colleague and some changes have been done.

It is true that the data can be vividly displayed in the form of a chart and it is also better to be visualized, but at the same time, detailed or precise data cannot be displayed. Moreover, there are 20 or 60 basis sets (or 10 ones in Table 1) for every method, their plots are too many (e.g., the 36 methods and 10 basis sets will lead to 360 data in Table 1) to be drawn in such narrow one page or several pages. In these tables, their MADs in different ranges (i.e., smaller than 0.010 Å, 0.010 Å~0.020 Å, 0.020 Å~0.1 Å, and larger than 0.1 Å) are clearly shown with different colors, which can make them more easily seen or read.